# Meta-QTL Analysis and Identification of Candidate Genes Associated with Stalk Lodging in Maize (*Zea mays* L.)

**DOI:** 10.3390/cimb47100792

**Published:** 2025-09-23

**Authors:** Haiyue Fang, Chunxiao Zhang, Wenli Qu, Xiaohui Zhou, Jing Dong, Xueyan Liu, Xiaohui Li, Fengxue Jin

**Affiliations:** 1Maize Research Institute, Jilin Academy of Agricultural Sciences (Northeast Agricultural Research Center of China), Gongzhuling 136100, China; fhy910326@163.com (H.F.); chunxiao1000@126.com (C.Z.); quwenli@163.com (W.Q.); zhouxiaohuijl@126.com (X.Z.); dongjing_1998@163.com (J.D.); xyliu@jaas.com.cn (X.L.); 2Qiqihar Sub-Academy of Heilongjiang Academy of Agricultural Sciences, Qiqihar 161006, China

**Keywords:** maize, stalk lodging, meta-QTL, candidate genes

## Abstract

Stalk lodging constitutes a primary constraint on achieving consistently high yields in maize. Genetic improvement of lodging resistance requires the identification of stable quantitative trait loci (QTL) to facilitate the application of genomics-assisted breeding for improving selection efficiency in breeding programs. In this study, we performed a meta-analysis to identify consensus loci and functionally characterized candidate genes associated with stalk lodging-related traits. Through meta-analysis integrating 889 reported lodging-related QTLs using the IBM2 2008 Neighbors high-density genetic map, we identified 67 meta-QTLs (MQTLs), of which 32 were determined as core MQTLs. Among them, 67% were validated by co-localized marker–trait associations from genome-wide association studies (GWAS). Comparative genomics further revealed 40 evolutionarily conserved orthologs via protein alignment with rice lodging genes, while screening of core MQTL regions detected 802 candidate genes with KEGG enrichment implicating galactose degradation II in cell wall reinforcement, supported by transcriptomic evidence of their roles in lignin biosynthesis pathways modulating mechanical strength. In conclusion, the MQTL identified and validated in our study have significant scope in marker-assisted selection (MAS) breeding and map-based cloning programs for improving maize stalk lodging.

## 1. Introduction

In recent years, escalating natural disasters, pest outbreaks, and altered planting regimes have exacerbated maize lodging risks globally. Global estimates indicate that stalk lodging causes annual yield losses of 5–20% [1], critically compromising production security. For sustainable maize production systems, developing lodging-resistant cultivars through breeding is increasingly recognized as a pivotal strategy to ensure high and stable yields [2,3]. Consequently, elucidating the genetic architecture underlying maize lodging resistance and mining functionally validated candidate genes are essential for precision breeding of lodging-resilient crops [4,5].

Maize lodging manifests as two distinct phenotypes: stalk lodging and root lodging [6], with the former typically causing more severe yield damage during equivalent growth stages [7]. As a polygenic quantitative trait, stalk lodging exhibits complex genetic architecture. QTL mapping enables the identification of genomic intervals controlling lodging resistance through molecular markers, facilitating marker-assisted selection (MAS) in breeding programs. Extensive QTL mapping studies have characterized lodging-related traits across multiple dimensions, including the following: (i) morphological parameters such as plant height, ear height, internode length, and stem diameter [8]; (ii) biochemical constituents encompassing cellulose, hemicellulose, lignin, and soluble sugar content [9,10]; (iii) biomechanical properties measured through rind penetrometer resistance and stalk bending strength [11]; and (iv) anatomical features including vascular bundle density and cross-sectional area [12]. However, substantial variations in genetic backgrounds, population structures, statistical methodologies, and testing environments, combined with limitations in marker density, frequently yield QTL confidence intervals exceeding 10 cM. This compromises the utility of QTL mapping for precision breeding of lodging-resistant maize.

Meta-analysis, employing mathematical models to integrate results from multiple studies, effectively narrows QTL confidence intervals (CI) while enhancing mapping precision and reliability [13]. This methodology has been successfully implemented in major crops including rice [14], maize [15,16], and soybean [17]. Current maize meta-analyses primarily address yield components [18], abiotic stress tolerance [19], and biotic resistance [20,21], with lodging-related traits remaining critically underrepresented. To address this gap, we conducted the first comprehensive meta-analysis of maize stalk lodging QTLs to establish a robust framework for candidate gene discovery and marker-assisted breeding.

In this study, we systematically curated documented QTLs associated with 12 key stalk lodging-related traits: plant height (PH), ear height (EH), rind penetrometer resistance (RPR), internode length (IL), stem diameter (SD), detergent fiber (DF), cellulose (Cel), hemicellulose (Hem), lignin (Lig), soluble sugar content (SSC), stalk bending strength (SBS), and vascular bundle characteristics (Vb). These QTLs were integrated via meta-analysis using a high-density consensus linkage map, generating a refined MQTL landscape for stalk lodging. Subsequent integration of MQTL coordinates with maize reference genome sequences (B73 RefGen_v4) and computational biology approaches enabled systematic candidate gene identification, collectively providing foundational resources for precision breeding initiatives.

## 2. Materials and Methods

### 2.1. Collection of Mapping and Quantitative Trait Locus (QTL) Data

QTL information for 12 stalk lodging-related traits including PH, EH, RPR, IL, SD, DF, Cel, Hem, Lig, SSC, SBS, and Vb was collected from 44 publications (2003–2024) [8,9,11,12,22,23,24,25,26,27,28,29,30,31,32,33,34,35,36,37,38,39,40,41,42,43,44,45,46,47,48,49,50,51,52,53,54,55,56,57,58,59,60,61] retrieved through databases including China National Knowledge Infrastructure (CNKI, https://www.cnki.net/, accessed on 1 December 2024) and the National Center for Biotechnology Information (NCBI, http://www.ncbi.nlm.nih.gov/, accessed on 1 December 2024) (Table 1). The compiled data encompassed the following: mapping population size, population type, mapping function, chromosomal position, LOD score, *R*^2^ value, genetic map, molecular markers, and genetic distance. These 12 traits were categorized into the following: stalk morphology (SM), stalk strength (SS), stalk chemical composition (SC), stalk anatomical structure (SA). All QTL data were derived from experiments conducted under normal conditions, excluding studies involving stress treatments. QTL position (maximum likelihood position and its confidence interval) and contribution rate (proportion of phenotypic variance explained) are two key parameters of QTL [62]. Where positional data was missing, the midpoint between flanking markers was substituted. For unknown confidence intervals, estimates were calculated using the following formulae [63,64,65]:C.I. = 530/(N × *R*^2^) (backcross and F_2_ populations) (1)C.I. = 163/(N × *R*^2^) (recombinant inbred line populations) (2)
where N represents population size and *R*^2^ denotes the QTL contribution rate. When LOD scores were unavailable, *R*^2^ was estimated using the following formula:
(3)$$R^{2}={1-10}^{\left(-2LOD/N\right)}$$


### 2.2. QTL Projection and Meta-QTL Analysis

The original QTL information collected in this study was derived from different mapping populations using varied genetic maps and molecular marker types, resulting in limited common markers available for constructing a consensus map. To address this, we utilized the high-density IBM2 2008 Neighbors genetic map (https://www.maizegdb.org/data_center/map, accessed on 20 December 2024) as a reference map. This comprehensive map incorporates diverse marker types and provides unambiguous locus order information for first-generation molecular markers (e.g., RFLP, RAPD) and second-generation markers (e.g., SSR). Meta-QTL analysis was performed using Biomercator v4.2 software [66]. The QTLs and common marker’ position on the reference map are computed by application of the appropriate distance ratio (homothetic projection), which is used to establish a linear, proportionally scaled positional mapping relationship between the common markers shared by two genetic maps. The absolute position of the QTL on the reference map is calculated by computing the confidence interval genetic distance ratio between the reference map and the original map, as well as the relative positional proportion of the QTL relative to the start of the confidence interval on the original map. This allowed the positions of QTLs from different sources, along with the coordinates defining the bounds of their confidence intervals, to be projected onto the IBM2 2008 Neighbors reference map. SNP markers were projected onto the reference map using markers with adjacent physical positions on the maize genome [14]. QTLs whose projected positions fell outside the range of the reference map were excluded.

Meta-QTL analysis employed the Veyrieras two-step algorithm [67]. Using a clustering method based on the Gaussian mixture model, this approach determines how many original QTLs underline the observed MQTLs. The parameters of this model were estimated via the EM algorithm. The QTL model estimated the number of MQTLs per chromosome using the five optional criteria: AIC (Akaike information criterion), AICc (AIC correction), AIC3 (AIC 3 candidate model), BIC (Bayesian information criterion), and AWE (average weight of evidence); we chose the results based on the AIC, which demonstrated strong performance in simulations, with the aim of achieving the comprehensive identification of all potential loci based on predictive accuracy [68]. The 95% confidence interval was used to estimate the interval of the MQTLs. The figures in this study were plotted using R.

### 2.3. Verification of MQTL with GWAS

To validate the identified MQTLs, marker–trait association (MTA) information from nine independent maize GWAS on lodging resistance was collected [58,69,70,71,72,73,74,75,76]. The physical positions of MTAs from these studies were compared with those of the MQTLs. The MTAs obtained from GWAS near MQTL (500 kb upstream and downstream of the MQTL regions) were considered as validation of the MQTL [77].

### 2.4. Identifying and Functional Annotation of Candidate Genes

To comprehensively identify and characterize candidate gene functions within MQTL regions, we referenced methods from previous studies [77]: (i) Query genes within the MQTL regions in the maize genome database (http://www.maizegdb.org/, accessed on 6 March 2025). The protein sequences of these genes were downloaded from the NCBI website (http://www.ncbi.nlm.nih.gov/, accessed on 6 March 2025) and their physical positions relative to the B73 reference genome were determined. Cloned rice lodging resistance genes were searched for using the China National Rice Data Center (https://www.ricedata.cn/, accessed on 1 April 2025), China National Knowledge Infrastructure (CNKI), and the National Center for Biotechnology Information (NCBI). BLASTP alignment was performed to compare the homology between the protein sequences of the rice genes and those of the genes within the MQTL regions. The alignment criterion was E < 1 × 10^−10^ and identity > 60%. (ii) Candidate genes within the breeders’ MQTLs [78] (comprising ≥2 original QTLs, <1 Mb physical size, <4 cM genetic size) were explored. (iii) We identified candidate genes located within the GWAS-MTA-validated MQTL. For MQTLs harboring >3 MTAs and a physical confidence interval longer than 1 Mb, we used a 1 Mb genomic region (500 kb flanking each side of the peak); peak physical positions of the MQTLs were calculated by Saini et al. [79]. To investigate the functions of candidate genes, we performed GO (Gene Ontology) and KEGG (Kyoto Encyclopedia of Genes and Genomes) analyses on candidate genes obtained through three methods, while GO enrichment analysis was conducted using the AgriGO v2 platform (http://systemsbiology.cau.edu.cn/agriGOv2/manual.php, accessed on 3 May 2025) to explore the biological functions of candidate genes. KEGG pathway analysis was performed using the Plant Reactome database (https://plantreactome.gramene.org/index.php?lang=en, accessed on 3 May 2025).

### 2.5. Analysis of Expression Patterns of Candidate Genes

The expression data (transcript profiling) for candidate genes were retrieved from “qteller” (https://qteller.maizegdb.org/, accessed on 6 March 2025) of maizeGDB using the B73 RefGen_v4 assembly [80]. The whole transcriptomic data encompasses 55 tissues/stages across multiple timepoints and tissues [81], and were available for the following: embryo (16, 18, 20, 22, 24 DAP), endosperm (16, 18, 20, 22, 24 DAP), internode (0, 6, 12, 18, 24, 30 NOPOL), internode (0, 6, 12, 18, 24, 30 POL), whole seed (2, 4, 6, 8, 10, 12, 14, 18, 20, 22, 24 DAP), anthers (R1), brace root (V13), crown root (V7, V13), cob (R1, V18), Silks (R1), Primary root (Z1, Z2, Z3, Z4), Stem (V1, V3), Tassel (V13, V18), Internode elongation zone (V5, V9), Whole primary root (7 d), Whole root system (3, 7 d). The expression level of candidate genes was evaluated by Fragments Per Kilobase of transcript per million mapped reads (FPKM) value, and the genes with FPKM value > 2 in tissue expression were screened for expression analysis [82], and log_2_ (FPKM + 1) was used for heatmap plotting.

## 3. Results

### 3.1. Genomic Distribution of QTLs Associated with Stalk Lodging-Related Traits in Maize Genome

A comprehensive dataset of 889 stalk lodging-related QTLs was compiled from 44 independent studies (2003–2024) [8,9,11,12,22,23,24,25,26,27,28,29,30,31,32,33,34,35,36,37,38,39,40,41,42,43,44,45,46,47,48,49,50,51,52,53,54,55,56,57,58,59,60,61] for meta-analysis (Appendix A; Figure 1). The collected QTLs were unevenly distributed across all 10 maize chromosomes, with plant height (PH) having the highest number (187), followed by rind penetrometer resistance (RPR) with 113 and lignin (Lig) with 110. Of the total QTLs, 835 (93.9%) were successfully projected onto the IBM2 2008 Neighbors reference map. The unmapped 54 QTLs (6.1%) were excluded due to their projected positions falling outside the reference map range. Original studies featured population sizes ranging from 118 to 1948 individuals, utilized predominantly SSR, RFLP, and SNP markers, and reported variable QTL densities (1–86 per study). The LOD scores of these QTLs ranged from 0.82 to 36.96, with distribution analysis revealing the following: 127 QTLs (14.3%) had scores < 3, 575 (64.7%) between 3–6, 130 (14.6%) between 6–9, 32 (3.6%) between 9–12, and 25 (2.8%) ≥ 12 (Figure 1a). Individual QTLs explained 1.3–43.0% of phenotypic variation (PVE/*R*^2^), with distribution analysis revealing the following: <5% (*n* = 150, 16.9%), 5–10% (*n* = 463, 52.1%), 10–15% (*n* = 173, 19.5%), 15–20% (*n* = 68, 7.6%), and ≥20% (*n* = 35, 3.9%) (Figure 1b). The predominance of minor-effect loci (68.95% with *R*^2^ < 10%) indicates polygenic control of stalk lodging traits. Twelve traits evaluated under standard conditions were functionally categorized into the following: SM (44.99%), SS (14.96%), SC (37.35%), and SA (2.7%) (Figure 1c).

### 3.2. Meta-Analysis of QTL for Stalk Lodging-Related Traits in Maize

Meta-analysis of the 835 successfully mapped QTLs identified 67 MQTLs, each integrating 1–54 original QTLs (Appendix A; Figure 2 and Appendix A). These 67 MQTLs were distributed across all 10 maize chromosomes, with chromosomal densities ranging from 3 MQTLs (chromosome 3) to 9 MQTLs (chromosomes 9 and 10).

MQTL confidence intervals spanned 0.06–18.63 cM, with an average of 4.86 cM. MQTL intervals containing a greater number of initial QTLs and traits across diverse genetic backgrounds and environments, are considered more reliable. Among these, MQTL3.3 encompassed 54 initial QTLs and 10 traits, MQTL5.5 encompassed 38 initial QTLs and 7 traits, and MQTL1.1 encompassed 33 initial QTLs and 9 traits. MQTL10.7 and MQTL10.8 both encompassed one initial QTL and one trait. A total of 18 MQTLs had physical intervals less than 1 Mb. Among these, 9 MQTLs had genetic distances less than 4 cM: MQTL1.1, MQTL1.8, MQTL2.5, MQTL2.7, MQTL3.3, MQTL5.5, MQTL6.6, MQTL7.5 and MQTL8.7, respectively.

### 3.3. Validation of MQTL via GWAS Co-Localization

To validate the meta-QTLs, we cross-referenced their physical positions with marker–trait associations (MTAs) from nine maize lodging resistance GWAS [58,69,70,71,72,73,74,75,76]. A total of 45 MQTL (upstream and downstream 500 kb regions) were found to overlap with 217 MTAs, which were identified in eight GWAS on lodging resistance in maize (Appendix A). Among them, MQTL9.4 overlapped with 36 MTAs and MQTL6.1 with 17 MTAs, followed by MQTL10.2 overlapping with 12 MTAs, MQTL7.6 and MQTL8.2 overlapping with 11 MTAs, respectively. Furthermore, there were 19 MQTLs overlapping with 3–9 MTAs and 13 MQTL overlapping with 2 MTAs, while 8 MQTLs only overlapped with 1 MTA.

### 3.4. Functional Characterization of Candidate Genes for Stalk Lodging Within MQTL Regions

Multiple genes regulating maize stalk lodging-related traits were localized within MQTL regions (Figure 2; Appendix A): *ZmRPH1* (MQTL1.1) encoding a microtubule-associated protein regulating plant and ear height [83]; *ZmCLA4* (MQTL4.2) acting as negative regulator of leaf angle through mRNA accumulation modulation, affecting stem gravitropism and cellular development [84]; *Bv1* (MQTL5.3) controlling auxin transport-mediated height regulation [85]; *ZmNST3* (MQTL6.1) as *NAC* transcription factor activating cell wall biosynthesis genes to enhance lodging resistance [86]; *DIL1* (MQTL6.4) encoding *AP2* transcription factor governing internode elongation [87]; *ZmNST4* (MQTL9.2) essential for secondary cell wall biosynthesis [88]; *DWARF3* (MQTL9.3) cytochrome *P450* enzyme in gibberellin-mediated height control [89]; *PHYB2* (MQTL9.8) modulating stem elongation [90]; *BHLH117* (MQTL10.3) phytochrome-interacting factor (*PIF*) regulating plant stature [91]; and *ZmYUC2* (MQTL10.4) mediating local auxin biosynthesis to modulate brace root angle and lodging resistance [92]. Additionally, cloned lodging resistance genes within ±1 Mb of MQTLs included *BRD1* (MQTL1.8), *ZmPHYC2* (MQTL5.1), *GA20ox5* (MQTL8.7), *ZmACS7* (MQTL10.8), *ZRP4* (MQTL6.6), and *RTH6* (MQTL1.3) [93,94,95,96,97,98].

To identify candidate genes regulating stalk lodging traits within MQTL regions, we implemented three complementary approaches. First, systematic screening of maize homologs for 280 rice lodging resistance genes identified 40 maize orthologs physically located within MQTL intervals through rigorous protein sequence alignment (Appendix A). Most of these orthologous genes in maize exhibit similar functions affecting stalk lodging-related traits. For example, *Zm00001d005775* (MQTL2.4) affects lodging-related traits similar to rice orthologous gene *OsCesA4*, with both being associated with plant height and stalk strength. *Zm00001d045463* (MQTL9.2) and its rice orthologous gene *OsSWN1* exhibit conserved functions, affecting internode length and lignin biosynthesis. These results indicate that the functions of these candidate genes are likely conserved in maize and rice. As a second approach, employing a breeder-oriented selection threshold (physical interval < 1 Mb, genetic distance < 4 cM, ≥2 initial QTLs) [99], we identified nine high-confidence MQTLs (MQTL1.1, 1.8, 2.5, 2.7, 3.3, 5.5, 6.6, 7.5, 8.7) yielding 149 candidate genes. For the third approach, prioritizing MQTLs co-localizing with >3 marker–trait associations (MTAs) revealed 24 MTA-MQTLs (MQTL1.4, 2.3, 2.4, 5.1, 5.2, 5.4, 6.1, 6.4, 6.5, 7.1, 7.5, 7.6, 8.1, 8.2, 8.6, 9.1, 9.2, 9.4, 9.6, 9.7, 9.9, 10.2, 10.4, 10.7), with the exclusion of overlapping MQTL7.5 and analysis of 19 large-interval (>1 Mb) within ±500 kb of peaks and four compact-interval (<1 Mb: MQTL5.1, 6.5, 7.1, 10.7) loci identifying 614 candidate genes; of the non-redundant integration of these nine breeder MQTLs and 24 MTA-MQTLs, 32 core MQTLs were identified (Figure 2). Core MQTLs are defined as meta-QTLs with physical intervals smaller than 1 Mb, containing at least two initial QTLs, and validated by independent GWAS signals, whose synthesis with maize–rice orthologs revealed 802 candidate genes regulating lodging resistance; we also established a comprehensive scoring system for these genes based on the four types: breeder’s MQTL localization (1 point), GWAS co-localization validation (1 point), rice ortholog with functional support (1 point), and expression in relevant tissues (1 point). Based on the accumulated evidence points, all 802 candidate genes were divided into three confidence tiers: high-confidence genes (3 points): 44 genes (5.49%); medium-confidence genes (2 points): 566 genes (70.57%); low-confidence genes (1 point): 192 genes (23.94%) (Appendix A).

### 3.5. Functional Annotation of Candidate Genes

We next performed GO and KEGG pathway enrichment analysis of the identified 802 candidate genes to determine their functional classification. Among them, 601 genes had GO annotation (Figure 3). The most abundant GO terms related to biological processes were cellular processes (GO:000998, 600/601, 99.83%), metabolic processes (GO:0008152, 564/601, 93.84%) and cellular metabolic processes (GO:0044237, 524/601, 87.19%). GO terms related to molecular function—binding (GO:0005488, 412/601, 68.55%), catalytic activity (GO:0003824, 377/601, 62.73%) and heterocyclic compound binding (GO:1901363, 247/601, 41.10%)—were also highly enriched. Cell part (GO:0044464, 456/601, 75.87%), cell (GO:0005623, 456/601, 75.87%) and intracellular (GO:0005622, 412/601, 68.55%) were enriched in the cellular component’s annotation. The KEGG metabolic pathway is significantly enriched with metabolism and regulation, mainly involving carbohydrate metabolism: galactose degradation II, cellulose biosynthesis, UDP-L-arabinose biosynthesis and transport, and UDP-D-xylose biosynthesis; hormone signaling, transport, and metabolism: Strigolactone signaling; and secondary metabolism: galactosylcyclitol biosynthesis, Myo-inositol biosynthesis, and Polyisoprenoid biosynthesis (Figure 4).

### 3.6. Expression Analysis of Identified Candidate Genes for Stalk Lodging in Maize

Expression patterns of candidate genes across developmental stages and tissues were analyzed using the qTeller database. Of 802 candidate genes, 601 exhibited detectable expression, with 406 showing high expression levels (Appendix A). Here, we focused on 189 candidate genes (40 maize–rice orthologous genes and 149 candidate genes within the breeders’ MQTL regions), of which 114 candidate genes with highly specific expression in various tissues were visualized (Appendix A; Figure 5). According to their different expression patterns, these 114 candidate genes were divided into three categories. In the first category, the expression levels were high in almost all tissues and stages. Among them, *Zm00001d017526* had the highest expression in all tissues and is involved in multiple stress responses; *Zm00001d038653*, *Zm00001d032346*, *Zm00001d021810*, *Zm00001d033186*, *Zm00001d021815*, *Zm00001d052494*, and *Zm00001d038658* were also highly expressed in all tissues. The second type was highly expressed only in some tissues: *Zm00001d038648* and *Zm00001d046112* are highly expressed in internode and root stages. The third type of gene was expressed in all tissues but the expression level was rather low.

## 4. Discussion

In recent decades, extensive QTL mapping studies have been conducted. However, the use of diverse genetic materials, population types, and statistical methods make it difficult to comprehensively detect all QTLs controlling target traits in individual mapping experiments. Due to limitations in marker density, the identified QTLs often exhibit large genetic distances in their confidence intervals, resulting in lower effectiveness. Consequently, these QTLs struggle to be effectively applied in breeding practices. Meta-QTL analysis can overcome these limitations by integrating QTL results obtained across different environments and genetic backgrounds to identify highly reliable consensus genomic regions. Khahani et al. [14] collected 1052 QTLs for yield-related traits to perform meta-analysis in rice and obtained 114 MQTLs with an average of 4.85 cM CI in the resulting MQTLs. Chen et al. [17] conducted a meta-analysis for quality-related traits with 1034 initial QTL in soybean, and 212 MQTLs were found. Karnatam et al. [15] also performed meta-analysis for root-related traits in maize with 917 original QTL and identified 68 MQTLs.

In this study, we integrated 889 stalk lodging-related QTLs using the IBM2 2008 Neighbors reference genetic linkage map for meta-analysis, identifying 67 consensus MQTL. Comparative analysis revealed 26 MQTLs overlapping with previous studies (Appendix A) [15,100], with most exhibiting significantly reduced confidence intervals. Notably, MQTL6.1 showed a highly consistent positional overlap across previous studies. Additionally, MQTL2.5, MQTL2.7, MQTL6.1, MQTL6.6, and MQTL8.6 had two overlapping intervals from previous studies. This indicates that multiple loci with distinct genetic effects collectively contribute to maize stalk lodging resistance. Furthermore, we identified nine breeder’s MQTLs with narrow physical intervals (<1 Mb) and 95% confidence intervals <4 cM. The genomic regions defined by these MQTLs represent high-confidence candidate regions for future gene cloning and serve as robust targets for marker-assisted selection in breeding for improved maize lodging resistance.

The integration of marker–trait associations (MTAs) from GWAS provides robust validation for MQTLs, reinforcing the stability and reliability of candidate genes within these genomic regions. Daryani et al. [101] found that a total of 52 MQTLs co-localized with 171 MTAs, while Li et al. [78] documented at least one MTA overlapping with 31 of 64 MQTLs. In our study, 45 MQTLs (within ±500 kb flanking regions) co-localized with 217 MTAs, validating 67.2% (45/67) of identified MQTLs through independent GWAS evidence. In particular, further characterization of these MQTLs will facilitate the identification of associated genes, playing a critical role in advancing our understanding of the molecular mechanisms underlying stalk lodging resistance in maize.

Through integration of three distinct methodologies, 802 candidate genes were identified within the MQTL intervals. GO enrichment analysis revealed statistically significant associations between enriched biological processes and maize lodging resistance. Notably, single-organism processes encompassing signal transduction and cell wall biosynthesis pathways were implicated [102]. Ions function as enzymatic cofactors in critical processes including cell wall lignification and signal transduction, directly influencing stalk structural integrity. The *OsGLR3.4*-mediated Ca^2+^ flux is essential for actin filament organization and vesicle trafficking; knockout of *OsGLR3.4* in rice results in brassinosteroid (*BR*)-regulated growth defects, reduced *BR* sensitivity, and dwarf phenotypes due to suppressed internode elongation [103]. Integral membrane components synergistically regulate cell wall biosynthesis, thereby modulating lodging resistance. In rice, *STRONG1* stabilizes cortical microtubules to guide cellulose synthase complex movement along microtubular trajectories, enabling transverse deposition of cellulose microfibrils that significantly enhance secondary wall thickness and mechanical resistance to lodging [104]. These findings collectively demonstrate that lodging resistance emerges through synergistic interactions of multiple mechanisms.

Further KEGG analysis indicated significant enrichment in metabolic and regulatory pathways. Research confirms that specific metabolic regulatory pathways substantially influence stalk strength. For instance, Strigolactones (*SLs*)—phytohormones involved in shoot branching suppression—interact with auxin signaling to regulate stem and root thickening [105]. Myo-inositol contributes directly to plant cell wall construction by providing precursors for pectin and hemicellulose synthesis, serving as a key determinant of stalk mechanical strength [106]. Cellulose biosynthesis, mediated by *RTH6*-encoded cellulose synthase, shapes maize root system architecture to enhance lodging resistance [98].

Galactose degradation II emerged as the most significantly enriched pathway (Figure 6), involving two key genes. *Zm00001d032346* (located in MQTL1.6) encodes UDP-glucose 4-epimerase (UGE), which catalyzes the conversion of UDP-galactose to UDP-glucose in galactose degradation II. Previous analysis demonstrated that *Zm00001d032346* (UGE4) represents a candidate gene potentially involved in ferulic acid metabolism, while diferulates (DFAs) promote cell wall cross-linking, reinforcing cell wall rigidity in maize [107,108]. As the ortholog of rice *Osfc24*, it encodes UGE catalyzing UDP-glucose (UDP-Glc) to UDP-galactose (UDP-Gal) conversion. UDP-Gal serves as substrate for carbohydrate, glycolipid, and glycoprotein biosynthesis. *Osfc24* mutants exhibit brittle leaves and internodes with reduced sclerenchyma thickness, altered cell wall composition, and disrupted cellulose microfibril orientation, collectively compromising mechanical strength [109]. *Zm00001d013245* (in MQTL5.1) encodes UDP-glucose 6-dehydrogenase (UGD), converting UDP-glucose to UDP-glucuronate. By analyzing the transcriptional regulatory network of GA-regulated lignin biosynthesis, *Zm00001d013245* is co-expressed with multiple genes related to cell wall synthesis and was significantly correlated with multiple lignin synthesis metabolites in maize [110], mirroring its rice ortholog *UGD4* (*Os12g0443500*) that enhances stress resistance through cell wall modification [111].

The second most enriched pathway was cellulose biosynthesis. Cellulose, a linear 1,4-linked homopolymer of β-D-glucopyranose, forms stable crystalline microfibrils through intermolecular interactions that provide structural support [112] and serves as the primary component conferring rigidity to plant cell walls [113]. Two key genes drive this pathway: *Zm00001d023810* (located in MQTL10.2) encodes cellulose synthase-like D1, which participates in cell wall polysaccharide formation during cell division [114]. This gene is orthologous to rice *DNL1* (encoding cellulose synthase-like D4) which regulates plant height [115]. *Zm00001d005775* (*CesA7*) encodes a cellulose synthase A catalytic subunit controlling plant height [116] orthologous to rice *OsCesA4* which participates in cellulose synthesis, forms part of the cellulose synthase complex for secondary cell wall formation, and critically influences stem strength [117].

The maize lodging resistance genes *ZmRPH1*, *ZmCLA4*, *BV1*, *ZmNST3*, *DIL1*, *ZmNST4*, *DWARF3*, *PHYB2*, *BHLH117*, and *ZmYUC2* have been identified in the MQTL regions [83,84,85,86,87,88,89,90,91,92]. Notably, *PHYB2* (located in MQTL9.8) encodes phytochrome B2, which regulates key aspects of seedling development such as mesocotyl elongation and chloroplast gene expression. This gene modulates multiple agronomic traits including plant height, ear height, stalk diameter, leaf sheath length, and internode length, while significantly influencing flowering time plasticity in maize. *BHLH117* (MQTL10.3) encodes a phytochrome-interacting factor governing plant height determination and salt stress tolerance, thereby impacting yield stability. *DIL1* (MQTL6.4) functions as an AP2 transcription factor that reduces plant height through internode length modulation. Additionally, it modifies leaf architecture to enhance high-density planting adaptability, contributing to yield improvement through optimized canopy structure.

This study leveraged the conserved genomic synteny between maize and rice (*Poaceae* family) to elucidate gene functions through ortholog analysis. Orthologous sequences revealed physiological function conservation, enhancing our understanding of maize gene networks. For instance, rice genes *DNL1* [115], *OsCesA4* [117], and *OsSWN1* [118]—known regulators of lodging resistance—exhibit parallel functions in maize, demonstrating the utility of cross-species homology for candidate gene identification. We identified 40 maize–rice orthologous lodging resistance genes within MQTL regions, indicating evolutionarily conserved trait regulation across gramineous species.

Tissue-specific expression profiling revealed 406 candidate genes exhibiting high tissue specificity in internodes, roots, and seeds, with 114 visualized (Figure 5; Appendix A). These genes showed pronounced expression in tissues functionally linked to stalk lodging resistance. For instance, *Zm00001d017526* (plasma membrane intrinsic protein PIP1b) displays ubiquitous strong expression across maize tissues. This gene mediates drought and salt stress responses while regulating plant growth; notably, knockout of its rice ortholog *OsPIP* reduces plant height [119]. Additionally, *Zm00001d046112* (spermine synthase) potentially regulates stalk sugar content, with its rice ortholog *OsSPMS1* encoding spermine synthase that negatively regulates cell expansion and significantly impacts stalk development [120]. Collectively, integrating maize–rice ortholog functional data with tissue expression patterns identified multiple high-confidence lodging resistance candidate genes within MQTL regions.

Meta-QTL analysis allows QTL detected from independent experiments to be grouped into classes, and a consensus estimation of QTL position to be made [13]. When collecting relevant QTL information, it is necessary to ensure accuracy, authenticity, and QTL numbers, because (i) differences in mapping population and meta-QTL analysis method can reduce precision in identifying MQTLs position; (ii) the ordering of the markers on some genetic maps can differ locally from their order on physical maps, leading to map errors, and (iii) MQTL analysis is less reliable with fewer QTLs per chromosome [68]. Although candidate genes were identified and functionally analyzed in this study, several limitations persist. The use of a ±500 kb window for GWAS co-localization, though commonly adopted, may not account for local variation in linkage disequilibrium patterns. Furthermore, while conserved orthologs provide functional clues, direct evidence of gene functions in maize stem strength is still limited for many candidates. Future work should focus on integrating multi-omics data to build regulatory networks for screening high-confidence candidate genes, followed by functional validation using genome editing techniques.

## Figures and Tables

**Figure 1 cimb-47-00792-f001:**
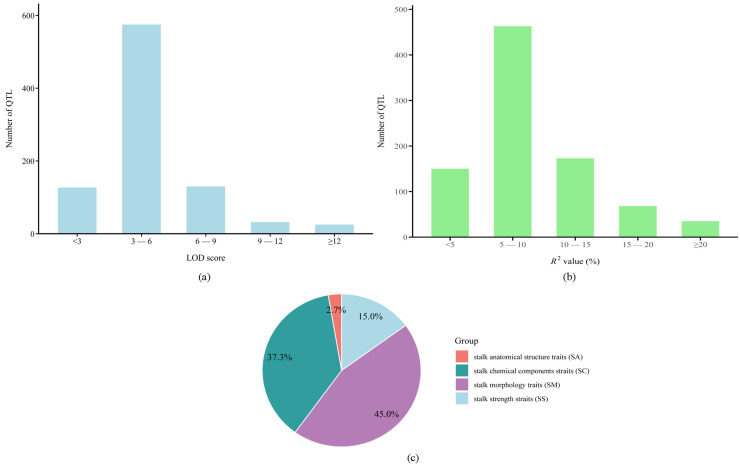
(**a**) Distribution of logarithm of odds (LOD) score, (**b**) distribution of *R*^2^ value or phenotypic variance explained (PVE), (**c**) classification of the traits. SA, stalk anatomical structure; SC, stalk chemical composition; SS, stalk strength.

**Figure 2 cimb-47-00792-f002:**
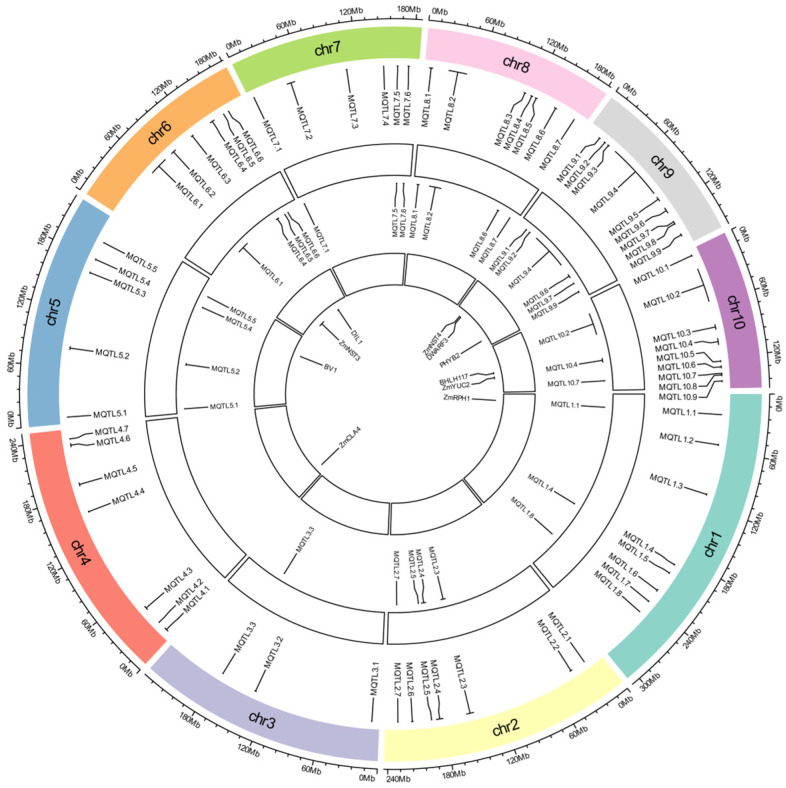
Circular plot of the stalk lodging-related MQTL in maize. From the inside to the outside: The innermost circle represents the cloned maize lodging resistance genes, the middle circle represents the physical intervals of core MQTL, and the outermost circle represents chromosomes with MQTL intervals.

**Figure 3 cimb-47-00792-f003:**
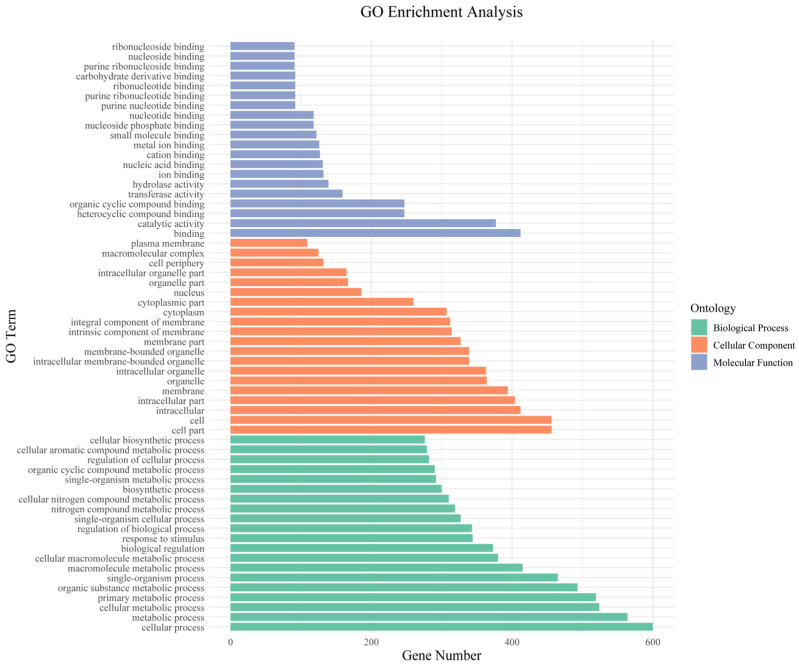
Enrichment of gene ontology (GO) terms for stalk lodging-related candidate genes identified in MQTL regions.

**Figure 4 cimb-47-00792-f004:**
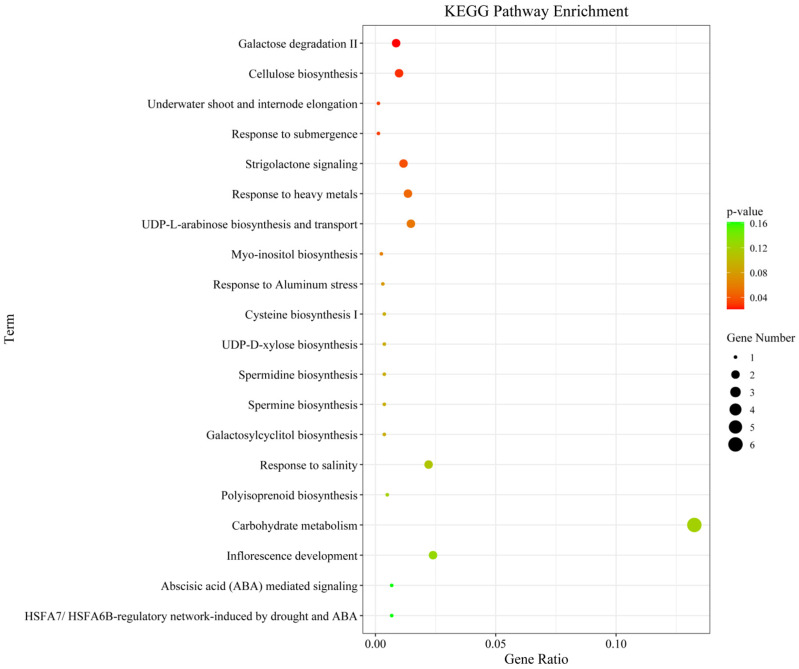
Top 20 KEGG enrichment pathways for the candidate genes identified for the MQTL regions.

**Figure 5 cimb-47-00792-f005:**
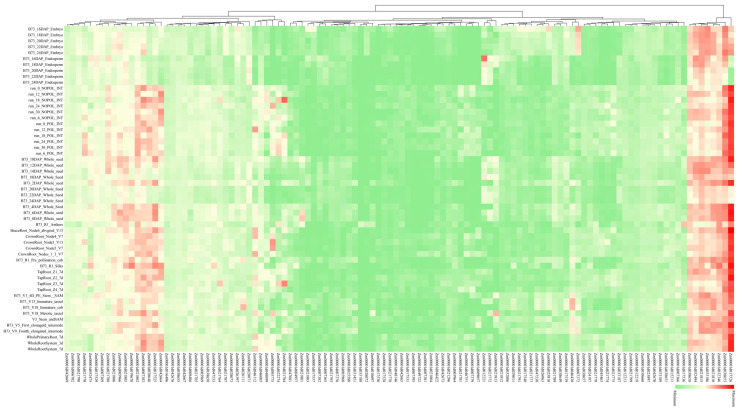
Heatmap of co-expression patterns for high-confidence lodging resistance candidate genes (≥2 FPKM) across diverse stem tissues in maize. Red represents high expression while green represents low expression.

**Figure 6 cimb-47-00792-f006:**
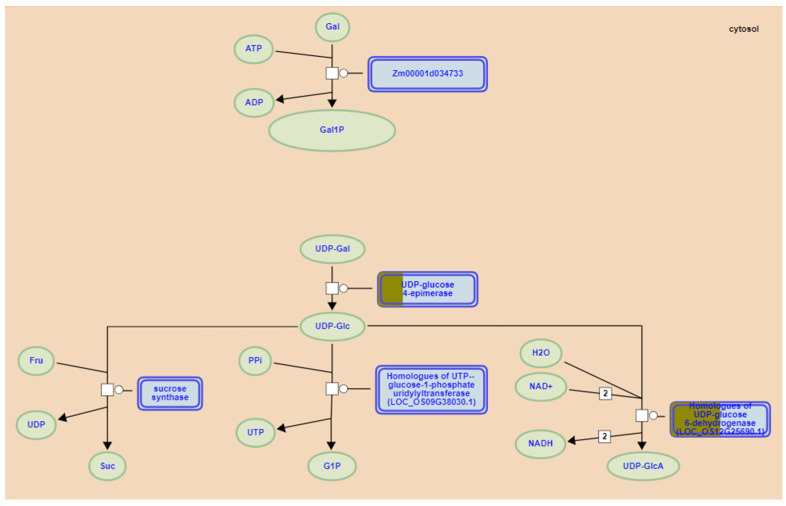
Regulation of Galactose degradation Il by lodging resistance candidate genes in maize (https://plantreactome.gramene.org/PathwayBrowser/#/R-ZMA-1119452 accessed on 3 May 2025). The brown in the blue box represents candidate genes.

**Table 1 cimb-47-00792-t001:** Collection of maize stalk lodging-related QTL information.

No.	References	Parents	QTL Number	Cross Type	Population Size	Trait	Marker Type	Method	Map Length (cM)	Map Density (cM)
1	Flint-Garcia et al. [22]	MoSQB-Low × MoSCSSS-Hight1	14	F2:3	282	RPR, EH	SSR	CIM	1468.5	17.27
		MoSCSSS-Hight2 × H25-LRP	16	F2:3	291	RPR, EH	SSR	CIM	1301.6	17.12
		Mo47 × MoSCSSS-Hight3	15	F2:3	291	RPR, EH	SSR	CIM	1353.3	16.7
2	Zheng and Liu [23]	Mo17 × Huangzao4	6	RIL	239	PH, EH	SSR	CIM	1421.5	14.2
3	Zhu et al. [8]	NX531 × NX110	17	DH	162	PH, EH, IL, SD	SSR	ICIM	1721.19	10.5
4	Ordás et al. [24]	EP42 × EP39	1	RIL	178	PH	SSR	CIM	1791	20
5	Yang et al. [25]	SICAU1212 × B73	6	F2	233	SD, PH, EH	SSR	ICIM	1290.4	11.53
6	Qiu et al. [26]	HZ32 × K12	4	F2	288	PH	SSR	CIM	1710.5	11.5
7	Fei et al. [27]	H132 × S122	11	F2:3	217	PH, EH, IL	SSR	ICIM	4734.51	27.69
8	Guo et al. [28]	5003 × p138	5	RIL	450	PH, EH	SSR	CIM	1395.2	13.81
9	Jiménez-Galindo et al. [29]	A637 × A509	3	RIL	171	PH	SNP	CIM	2372.1	8.6
10	Osman et al. [30]	HZ32 × K12	3	F2	247	PH	SSR	ICIM	1826.4	8.15
11	Li et al. [31]	B73 × By804	3	RIL	200	RPR	SNP	CIM	1600.40	2.1
		H127R × Chang7-2	4	RIL	215	RPR	SNP	CIM	1397.10	1.7
12	Du et al. [32]	(Ye478 × Zheng58) × Ye478	8	DH	123	PH, EH	SNP	CIM	1479.4	1.36
		(Ye478 × Zheng58) × Zheng58	10	DH	163	PH, EH	SNP	CIM	1872.1	1.44
13	Ku et al. [33]	Yu82 × Yu87-1	16	RIL	208	IL	SNP	CIM	1873.02	1.59
		Yu82 × Shen137	13	RIL	197	IL	SNP	CIM	1839.75	1.65
		Zong3 × Yu87-1	25	RIL	223	IL	SNP	CIM	1863	1.5
		Yu537A × Shen137	15	RIL	212	IL	SNP	CIM	1629.48	1.48
14	Luo et al. [34]	PH6WC × PH4CV	8	DH	240	PH	SNP	CIM	1462.05	1.11
15	Lima et al. [35]	L-20-01F × L-02-03D	8	F2:3	256	PH, EH	SSR	CIM	1858.61	13.47
16	Tang et al. [36]	Z3 × 87-1	21	RIL	294	PH	SSR	CIM	2361	9
		Z3 × 87-1	25	IF2	441	PH	SSR	CIM	2361	9
17	Wang et al. [37]	Zheng58 × W499	4	DH	118	EH	SNP	ICIM	2152.45	2
18	Zhang et al. [38]	KUI3 × B77	29	RIL	177	PH, EH	SNP	CIM	1640.4	0.74
19	Li et a. [39]	Zheng58 × Chang7-2	9	F2:3	225	PH, EH	SSR	CIM	1987.7	11
		BT-1 × N6	12	RIL	250	PH, EH	SSR	CIM	1820.8	11.7
20	Hou et al. [40]	37,051 × LH277	1	F2	214	RPR	SSR	ICIM	1312.00	5.11
			10	F2:3	214	RPR	SSR	ICIM	1312.00	5.11
21	Yan et al. [41]	Zong 3 × 87-1	52	F2:3	266	PH	SSR, RFLP	CIM	2531.6	14.5
22	Meng et al. [42]	Zheng58 × Chang7-2	5	DH	190	RPR	SNP	CIM	1426.83	1.26
23	Lemmon and Doebley [43]	teosinte × W22	3	BC6S6	259	PH, SD	RFLP	—	86.64	3.46
24	Zhang et al. [44]	Zheng58 × B73	11	F3:4	165	PH, EH	SSR	IM	2058.8	10.89
25	Cardinal et al. [45]	B52 × B73	34	RIL	200	DF, Lig	RFLP, SSR	CIM	1621	10.8
26	Barrière et al. [46]	F286 × F838	15	RIL	242	Lig	SSR	CIM	1911.2	17.2
27	Penning et al. [47]	B73 × Mo17	18	RIL	263	Lig, SSC	SNP	CIM	3262.5	1.5
28	Courtial et al. [48]	F288 × F271	21	RIL	131	Lig	SSR	CIM	2153.2	11.7
29	Wang et al. [49]	Ce03005 × B73	14	F3, F4	211	DF	SSR	CIM	2277.8	17.9
30	Hu et al. [50]	B73 × Ce03005	5	RIL	216	Cel, Lig, SBS	SSR	CIM	2016.52	15.6
31	Krakowsky et al. [51]	B73 × De811	44	RIL	191	DF, Lig	RFLP, SSR	—	1551	11.2
32	Virlouvet et al. [52]	F271 × Cm484	28	RIL	267	Lig, SSC, Cel, Hem	SNP	—	2355	2.4
33	Lorenzana et al. [53]	B73 × Mo17	86	RIL	223	Lig, SSC	RFLP, SSR	CIM	6240	4.7
34	Wei et al. [54]	GY220 × 8984	19	F2:3	284	DF, Hem	SSR	CIM	2111.7	11.41
		GY220 × 8622	12	F2:3	265	DF, Hem	SSR	CIM	2298.5	13.29
35	Li et al. [55]	Zheng58 × HD568	11	RIL	220	DF, Cel, Lig	SNP	CIM	1985.6	1.5
36	Barrière et al. [9]	RIo × WM13	28	RIL	163	Cel, Hem, Lig	SSR	CIM	1469	15
37	Zhang et al. [11]	B73 × Mo17	56	DH	221	RPR, PD, SBS	SNP	CIM	1767.45	0.29
38	He et al. [56]	Xu178 × K12	18	RIL	150	PH, EH	SSR	ICIM	2069.1	10.8
39	Yu et al. [57]	KA105 × KB020	21	F5	201	RPR, PH, EH	SNP	ICIM	—	—
40	Zhao et al. [58]	ROAM	25	RIL	1948	RPR	SNP	CIM	—	—
41	Huang et al. [12]	W22 × 8759	1	RIL	866	Vb	SNP	—	—	—
42	Ye et al. [59]	Y915 × Zheng58	13	RIL	171	SD, RPR	SNP	CIM	—	—
43	Zhang et al. [60]	hengbai522 × tongxi5	4	RIL	198	RPR	SNP	—	—	—
44	Du Yuxi [61]	X178 × NX531	23	RIL	248	Vb	SNP	CIM	2569	0.35

## Data Availability

All relevant data are within the paper and its Appendix A.

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
