# Peer review of "Meta-QTL Analysis and Identification of Candidate Genes Associated with Stalk Lodging in Maize (Zea mays L.)"

_cimb, 2025, doi:10.3390/cimb47100792_

Round 1
Reviewer 1 Report
Comments and Suggestions for Authors
In the current study, the authors analyzed QTL results from a collection of published work and correlated to stock lodging-related traits. A major portion of the identified core MQTLs were then validated by corroborating GWAS data. Genes of interest that are orthologues to those related to rice lodging were identified. Lastly, KEGG enrichment analysis highlighted galactose degradation II, which accounts for cell wall strength.
A few suggestions:
- It would be helpful to elaborate on the 44 studies from which the core source data were acquired. A few sentences summarizing the general subjects of the studies, the potential variations in genetic backgrounds, etc., would suffice.
- Please furnish the Supplementary information in the revised manuscript.
- Figures 1, 3, 4, 5, 6 and 7: please expand the figure legends to help the readers better understand the contents in the figures.
- Many characters in figure 1, 2, 3 and 6 are not easily recognizable, please improve the resolution of the images.
- Figure 3 appears a bit cramped, it may be helpful to move it to supplementary figures and/or keep the maps for just one or two chromosomes.
- Please fix the typo in the second sentence in Introduction.
- Please fix the font in the paragraph under “2.3. Verification of MQTL with GWAS Studies”, and the third paragraph in Discussion (“Daryani et al. [104] found that a total of 52 MQTLs colocalized with 171 MTAs”).

Reviewer 2 Report
Comments and Suggestions for Authors
I have gone through the manuscript titled, “Meta-QTL Analysis and Identification of Candidate Genes Associated with Stalk Lodging in Maize (Zea mays L.), where the authors presented a comprehensive meta-analysis of QTLs associated with stalk lodging in maize. The integration of comparative genomics with rice, functional enrichment analysis, and expression profiling provides valuable multi-evidence support for the candidate genes. The topic is of significant importance for maize breeding, and the work is largely well-conceived. However, several major concerns must be addressed before the manuscript can be considered for publication.
- Authors should consider briefly defining "core MQTL" when first mentioned in the introduction or results for clarity.
- I think, the description of the homothetic function used to project QTLs from various original maps onto the IBM2 2008 Neighbors reference map is insufficient. Please provide a more detailed explanation or a reference to the specific mathematical procedure. A supplemental table showing a sample of original QTL positions, their flanking markers, and their projected positions would greatly enhance transparency.
- The manuscript states the use of the Veyrieras algorithm and the minimum AIC value to determine the number of MQTLs. Please specify the AIC threshold or range of values used for model selection. Furthermore, for the "breeders MQTL" criteria (physical size <1 Mb, genetic size <4 cM, ≥2 initial QTLs), please justify the choice of these specific thresholds with references or rationale, as they are critical for defining the high-confidence set.
- Figures 1, 2, 3, 6, and 7 are referenced but are of low resolution and lack clear legends in the text provided. High-resolution versions with detailed captions are suggested to add. Especially, Figure 2 (Circular plot) and Figure 3 (QTL distribution) need to be legible. Figure 6 (heatmap) also requires clear sample/tissue labels and a color scale.
- In the manuscript, authors used a ±500 kb window for GWAS co-localization that is previously used by some studies but somewhat arbitrary. It is suggested to discuss the limitations of this approach. Were there any statistical measures used like, LD decay in the studied populations used to inform this window size?
- Even Hough three methods are used to identify candidate genes, the evidence for many remains preliminary being based on positional and homology evidence. The authors are advised to add a ranked list or a classification with high, medium, low confidence for the 802 candidate genes based on the strength of evidence that may be presence in a breeders MQTL + GWAS validation + rice ortholog + expression in relevant tissues.
- While KEGG analysis highlights "Galactose degradation II" as the most enriched pathway. The discussion of the two key genes (UGEand UGD) is good. However, please elaborate on the direct mechanistic link between this pathway and lignin biosynthesis/cell wall reinforcement in maize, moving beyond the analogy to rice. Did you find any direct transcriptional or metabolic evidence in maize?
- In results the sentence, "The unmapped 54 QTLs (6.1%) were excluded probably due to chromosomal inversions..." is speculative. Rephrase to state they were excluded because their projected positions fell outside the reference map range, as correctly stated in the Methods section.
- I feel the discussion heavily focuses on the successes of the study ignoring potential limitations. It is advised to add a paragraph dedicated to the limitationsof the study.
This study has the potential to be a significant contribution to the field of maize genetics and breeding for lodging resistance. Although, scope of the meta-analysis is impressive, listed concerns must be thoroughly addressed before the manuscript is formally accepted for publication in CIMB.

Round 2
Reviewer 2 Report
Comments and Suggestions for Authors
The revisions are thorough and address all major and minor concerns raised in comments. The manuscript now meets the expected standards for publication.